# Quality of Life of Cancer Patients Treated with Chemotherapy

**DOI:** 10.3390/ijerph17196938

**Published:** 2020-09-23

**Authors:** Anna Lewandowska, Grzegorz Rudzki, Tomasz Lewandowski, Michał Próchnicki, Sławomir Rudzki, Barbara Laskowska, Joanna Brudniak

**Affiliations:** 1Institute of Healthcare, State School of Technology and Economics in Jaroslaw, 37-500 Jaroslaw, Poland; barbara.laskowska917@gmail.com (B.L.); awelc@poczta.fm (J.B.); 2Chair and Department of Endocrinology, Medical University of Lublin, 20-090 Lublin, Poland; grzegorz.rudzki@orange.pl; 3Institute of Technical Engineering, State School of Technology and Economics in Jaroslaw, 37-500 Jaroslaw, Poland; tom_lew@interia.pl; 4I Department of Psychiatry, Psychotherapy and Early Intervention, Medical University of Lublin, 20-439 Lublin, Poland; michal.prochnicki@umlub.pl; 5I Chair and Department of General and Transplant Surgery and Nutritional, Medical University of Lublin, 20-090 Lublin, Poland; slawomir.rudzki@umlub.pl

**Keywords:** cancer, quality of life, chemotherapy

## Abstract

*Background:* Life-quality tests are the basis for assessing the condition of oncological patients. They allow for obtaining valuable information from the patients regarding not only the symptoms of disease and adverse effects of the treatment but also assessment of the psychological, social and spiritual aspects. Taking into account assessment of the quality of life made by the patient in the course of disease has a positive effect on the well-being of patients, their families and their caregivers as well as on satisfaction with the interdisciplinary and holistic oncological care. *Methods*: A population-based, multi-area cross-sectional study was conducted among patients with cancer in the study in order to assess their life quality. The method used in the study was a clinical interview. Quality of life was measured using the EQ-5D-5L Quality of Life Questionnaire, the Karnofsky Performance Status, our own symptom checklist, Edmonton Symptom Assessment and Visual Analogue Scale. *Results:* In the subjective assessment of fitness, after using the Karnofsky fitness index, it was shown that 28% (95% CI (confidence interval): 27–30) of patients declared the ability to perform normal physical activity. In the assessment the profile, quality of life and psychometric properties of EQ-5D-5L, it was shown that patients had the most severe problems in terms of self-care (81%, 95% CI: 76–89) and feeling anxious and depressed (63%, 95% CI: 60–68). *Conclusions:* Cancer undoubtedly has a negative impact on the quality of life of patients, which is related to the disease process itself, the treatment used and the duration of the disease.

## 1. Introduction

Contemporary oncology focuses not only on pharmacological treatment but also on a fuller understanding of the experiences of patients and their families, prioritizing the allocation of resources and planning and providing holistic care that will measurably affect the quality of life [1,2]. The quality of life in cancer is a dynamic, multidimensional concept, referring to all life aspects and needs of the patient, continuously assessing balancing processes between the real situation and the ideal situation at a given time [3,4,5]. Quality of life is a subjective feeling, mostly determined by individual needs, beliefs, values and attitudes; moreover, it is a value that changes over time [6]. An analysis of the literature shows that cancer patients can have very different categories of needs. The disease is an unpleasant experience to which every person reacts individually. Mental reactions related to disease and the patient’s needs depend on personality traits and the patient’s understanding of their new situation. The mental state of the patient changes with time, disease progression and treatment, and a positive attitude plays an important role in the recovery process [7,8,9].

Quality of life is usually assessed subjectively by the patient, and if this is not possible, assessment may be made by a doctor, nurse or caregiver. Life-quality tests are the basis for assessing the condition of oncological patients. They allow for obtaining valuable information from the patients regarding not only the symptoms of disease and adverse effects of the treatment but also assessment of the psychological, social and spiritual aspects. Taking into account assessment of the quality of life made by the patient in the course of disease has a positive effect on the well-being of patients, their families and their caregivers as well as on satisfaction with the interdisciplinary and holistic oncological care [10,11,12,13].

### Objective of the Work

The aim of the study was to assess the quality of life of cancer patients during oncological chemotherapy treatment.

## 2. Data and Method

### 2.1. Study Design

A population-based, multi-area cross-sectional study was conducted among patients of the Podkarpackie Oncology Centre, Clinical Provincial Hospital in Rzeszów in 2018–2020. Patients diagnosed with cancer were invited to participate in the study in order to assess their life quality. Due to the small sample size, it was important to include patients with fairly consistent characteristics. 

Eligible patients were given an information package by their medical specialist, who was part of the research group. The information package consisted of a letter that outlined the objectives of the study and what participation would involve, a consent-to-contact form to be completed if patients were interested in the study and a non-consent sheet. After informed consent was obtained, an interview was conducted at the clinic by an interviewer who was part of the research group. The interview lasted approximately 40 min. In the case of patient fatigue, the interview was divided into parts in order to maintain the physical and mental comfort of the patients.

#### Participant Recruitment, and Inclusion and Exclusion Criteria

Eligible patients were identified from the department of oncology within three months of diagnosis. Medical specialists who were part of the research group recruited patients for the study. The main indicators of participation in the study were the diagnosis of cancer at least three months before the study, life expectancy over 6 months, age over 18 years, chemotherapy treatment, no history of other chronic diseases and awareness of the diagnosis. We recruited only patients with solid cancer since patients with haematological malignancies tend to have a very different quality of life trajectory as well as prognosis compared to those with solid tumours. We excluded from the study patients whose cancer diagnosis was shorter than three months because the initial period of diagnosis and treatment is associated with a huge psychological burden and the need to adapt to the new situation, which may cause errors in the results. Patients who were too physically ill, too emotionally distressed, <18 years of age or not literate in Polish were excluded. To avoid obtaining a very heterogeneous sample, we also excluded those on hospice or home-based care or those taking only palliative care treatment for more than 2 weeks.

After signing the informed consent form, information was obtained from patients regarding their sociodemographic characteristics, family history of cancer, first symptoms attributable to cancer perceived by the patient, symptom perception and reaction to early symptoms. At the same time, patients were asked to complete the questionnaire concerning their health-related quality of life. Quality of life was measured using the EQ-5D-5L Quality of Life Questionnaire, the Karnofsky Performance Status, our own symptom checklist, Edmonton Symptom Assessment and Visual Analogue Scale. In the case of patients qualifying for curative surgical treatment, the initial interview was performed in the month following discharge after surgery. Clinical records were also reviewed to collect information about comorbidity at diagnosis, tumour characteristics at diagnosis, treatment received, hospital consultations and exploratory procedures at follow-up.

### 2.2. Sample

The study included 800 patients, with 60% women and 40% men diagnosed with cancer, undergoing chemotherapy treatment; 58% of rural residents and 42% of city residents participated in the study. The average age of the patients was SD 54.52 (8.86) years. All patients were treated as intended by the public health system.

#### Questionnaire for the Patient

The method used in the study was a clinical, direct, individual, structured interview, which was in-depth and focused. The qualitative interview questionnaire was a standardized measuring instrument and was verified by testing a group of 30 patients during the month. The questionnaire has acceptable levels of internal consistency and test–retest reliability as well as construct validity. It contained detailed and extensive instruction for the interview. The questionnaire was approved by a clinical psychologist employed at the oncology clinic. It contained open-ended, single-choice and multiple-choice questions, allowing to obtain recorded and epidemiological information as well as to assess patient’s general condition, psychological, physical, everyday-life, sexual and spiritual needs, as well as care needs.

### 2.3. Data on Quality of Life

Quality of life was measured using the EQ-5D-5L Quality of Life Questionnaire, the Karnofsky Performance Status, our own symptom checklist, Edmonton Symptom Assessment and Visual Analogue Scale.

The EQ-5D-5L Quality of Life Assessment questionnaire includes questions about walking ability, self-care, daily activity, pain and discomfort, anxiety and depression. The questions relate to the day on which the questionnaire was completed by the patient; one of 5 responses was selected for each scale.

The Karnofsky Performance Status (KPS) is a method for determining physical functioning using a metric assessment of the degree of functional independence expressed as a single number. This assessment is placed on a scale from 0 to 100: 0 meaning death to 100 meaning fully active. It is assumed that higher scale values correspond to better fitness and a higher quality of life. The results obtained with this scale in relation to oncological patients highly correlate with the survival time.

The symptom checklist is used to measure the impact of cancer treatment on the occurrence of psychological and physical symptoms. The psychological factor includes the following symptoms: nervousness, depression, difficulty sleeping, anxiety and difficulty concentrating. The physical factor includes the following symptoms: loss of appetite, fatigue, nausea, pain and hair loss.

The Edmonton Symptom Assessment System (ESAS) is a simple, important and reliable tool designed to assess the quality of life of cancer patients. It includes visual-analogue scales to assess symptoms: pain, activity, nausea, anorexia, well-being, dyspnoea, depressed mood, anxiety, vomiting and constipation. All items are scored on an 11-point scale, with 0 being no symptoms and 10 being the highest severity of the symptom.

The Visual Analogue Scale (VAS) is a reliable tool used to measure the severity of pain. Cyclically repeated measurements of pain intensity using the VAS scale enable the assessment of the effectiveness of analgesic treatment. The scale is a 10-cm ruler where 0 is no pain at all, 1–3 is mild pain, 4–6 is moderate and 7–10 is severe.

#### Ethical Considerations

The study was approved by the Bioethics Committee at the University of Rzeszów (Resolution No. 2017/12/4). Participation in the study was voluntary and anonymous, and respondents were informed of their right to refuse or withdraw from the study at any time. Each participant was informed about the purpose of the study and the time of completion of the study.

### 2.4. Data Analysis

All data obtained were collected and analysed with Prism 4.0 software. Descriptive statistics and confidence intervals were used to analyse participant characteristics, demographics and prevalence of needs. Statistical characteristics of continuous variables are presented in the form of arithmetic means, standard deviations and medians. Statistical characteristics of step and qualitative variables were presented in the form of numerical and percentage distributions. Cronbach’s α composite scales and the subscales were used to assess internal consistency. The relations between unmet needs and quality of life were analysed using linear regression. The correlations were determined using the Pearson test, while χ^2^ was used for the comparison between the groups. Significance was assessed at the level of *p* < 0.05. The repeatability of answers to individual questions was assessed using Kappa Cohen statistics. Missing data were excluded from all analyses.

## 3. Results

### 3.1. Baseline Demographics

The study group consisted of 60% women and 40% men. The mean age of the respondents was SD 54.52 (8.86). The youngest person was 26 years old, and the oldest was 75 years old. Table 1 presents other descriptive statistics identifying the studied group.

Of the respondents, 35% (95% CI: 32–39) declared they detected symptoms of cancer on their own, 33% (95% CI: 29–37) had their symptoms detected by a doctor and 13% (95% CI: 10–19) declared accidental detection of the disease. Prophylactic examinations contributed to detection of cancer in 19% (95% CI: 12–23) of the respondents. The reasons for visiting a doctor were nodules and lumps for 28% (95% CI: 22–31) of respondents, pain for 23% (95% CI: 20–29), a chronic weakness for 15% (95% CI: 12–19), bleeding for 20% (95% CI: 15–23) and increased temperature for 5%. Some patients declared that they had not experienced any disturbing symptoms (15%, (95% CI: 12–19). According to the results of a simple analysis of the level of a cancer diagnosis for the use of prophylactic examinations and self-observation, there was a statistically significant relation to sex and education. Women and better-educated people used prophylactic examinations more often and paid more attention to disturbing symptoms (*p* = 0.05). Among the factors that may have contributed to disease, patients listed successively genetic factors (70%, 95% CI: 68–73), smoking (51%, 95% CI: 46–59), improper diet (19%, 95% CI: 15–26) alcohol (17%, 95% CI: 15–24), chemical compounds (11%, 95% CI: 8–17), numerous sexual relations (8%, 95% CI: 2–12), viruses (6%, 95% CI: 2–12) and bacteria (4%, 95% CI: 2–12). During the assessment of occurrence of cancer in the families of the respondents, it was shown that the most frequently indicated occurrence of cancer was breast cancer among subjects’ mothers (30%, 95% CI: 25–35), while among their fathers, it was lung (19%, 95% CI: 13–23) and colorectal cancer (15%, 95% CI: 13–23).

#### 3.1.1. Psychological Impact—Questionnaire and Wellbeing

Of the patients, 38% (95% CI: 32–39) reacted with shock to their diagnosis, 37% (95% CI: 32–39) could not believe in the diagnosis, 22% (95% CI: 20–25) had a breakdown due to the received news and 3% (95% CI: 2–6) felt helpless; 48% (95% CI: 39–51) used the help of a psychologist during their illness, and 10% (95% CI: 9–13) used the help of a psychiatrist. During the disease, patients received support mostly from their closest relatives: spouse (62%, 95% CI: 59–69) and children (22%, 95% CI: 19–29). Support received from medical and nursing staff was appreciated by 17% (95% CI: 12–19) of patients. According to the results of a simple analysis of the level of frequency of using professional help and the support received, there was a statistically significant relation to the sex of the respondents. Women significantly more often than men used the professional help of a psychologist or psychiatrist and received support from their children. The differences were statistically significant (*p* = 0.03). The duration of disease did not significantly affect feelings of sadness and depression in the studied patients (*p* = 0.5). More than half of the respondents (65%) declared that they could cope with cancer, and 42% (95% CI: 38–49) accepted the disease. The vast majority of respondents said that they were motivated to fight the disease (57%, 95% CI: 51–60). Despite illness, 22% (95% CI: 18–27) of patients declared enjoying life, 13% (95% CI: 10–17) had lowered self-esteem, 8% (95% CI: 3–9) felt guilt, 35% (95% CI: 32–39) felt anger, 12% (95% CI: 8–16) felt ashamed of cancer and 30% (95% CI: 28–39) felt depressed. Almost half (47%, 95% CI: 42–49) of the respondents admitted to feeling more anxious. The vast majority of respondents did not blame anybody for their illness. Patients who felt such resentment named their doctor (12%, 95% CI: 8–15), family (3%, 95% CI: 1–6), themselves (3%, 95% CI: 1–6) and God (16%, 95% CI: 10–21). Most respondents (87%, 95% CI: 81–93) had positive thoughts about their lives, but only 30% (95% CI: 28–36) of patients had plans for the future. As many as 48% (95% CI: 42–51) feared for their future. According to the results of a simple analysis of the level of frequency of coping with the disease and motivation to fight the disease, there was a statistically significant relation to the sex of the respondents and the stage of the disease. Women more often than men declared being motivated and coping with the disease. Patients suffering for more than 3 years had less motivation to fight the disease and more often felt fear of the future. The differences were statistically significant (*p* = 0.01). In the course of statistical analyses, it was noticed that the duration of disease significantly influenced assessment of the quality of life of the studied patients (*p* = 0.01), as opposed to their age (*p* = 0.53). The duration of the disease did not significantly affect feelings of depression in the studied patients (*p* = 0.05).

#### 3.1.2. Impact on Social Interactions

In the vast majority of respondents (37%, 95% CI: 31–42), relations with family did not change due to the disease and they were good, and 15% (95% CI: 11–18) of respondents defined them as very good and even better than before. Some patients claimed that it was the disease that made them closer to their family and friends (15%, 95% CI: 8–19). On the other hand, growing distance from their family affected 20% (95% CI: 14–22) of respondents. For 13% (95% CI: 8–16), no changes within relationships occurred. During the disease, relationships with partners changed for the better in 28% (95% CI: 18–30) of patients, deteriorated in 23% (95% CI: 18–26) and remained unchanged in 49% (95% CI: 43–50). The research showed that 46% (95% CI: 46–50) of surveyed patients limited social contacts due to cancer, while 54% (95% CI: 53–55) of patients declared the opposite. When asked about the values important to them before their illness, the patients mentioned family (90%, 95% CI: 88–92), health (72%, 95% CI: 70–73), work (67%, 95% CI: 65–70), money (47%, 95% CI: 44–49), respect (35%, 95% CI: 33–37) and trust (27%, 95% CI: 26–29). Values that were important for the respondents during their illness were health (87%, 95% CI: 86–89), family (57%, 95% CI: 55–59), money (17%, 95% CI: 16–19), trust (17%, 95% CI: 16–19) and respect (10%, 95% CI: 6–14). Relationships with family and friends were not significantly related to the age, sex or the place of residence of the respondents, but the relation of the duration of the disease to relationships with children was confirmed (*p* = 0.01).

#### 3.1.3. Symptoms

When assessing well-being during treatment, the majority of patients (72%, 95% CI: 70–77) stated that they felt well, while 28% (95% CI: 21–31) of patients declared that they felt unwell. Most of the patients (42%, 95% CI: 36–49) assessed their health as average, 37% (95% CI: 32–40) as good and 22% (95% CI: 18–26) as poor; 28% (95% CI: 23–32) of patients experienced high stress related to the disease. When asked if they felt attractive and satisfied with their appearance, the vast majority of patients responded negatively (71%, 95% CI: 69–79). Only 40% (95% CI: 38–46) of patients used various forms of improving their appearance. The vast majority of respondents (65%, 95% CI: 63–69) admitted that they were not satisfied with their sex life. According to the results of a simple analysis of the level of self-assessment of attractiveness and the use of various methods of improving one’s appearance, there was a statistically significant relationship between the sex of the respondents and the duration of illness. Women more often than men felt unattractive and more often used the opportunity to improve their appearance. Patients who were ill for more than 3 years often felt unattractive. The differences were statistically significant (*p* = 0.03).

During assessment of the symptoms, the most frequently reported symptoms were fatigue, lack of appetite, difficulty sleeping, constipation and pain (Table 2). By assessing the severity of individual ailments according to the ESAS scale, it was shown that the highest severity of symptoms was related to constipation and fatigue. In assessing pain intensity on the VAS scale, the mean was 73% (95% CI: 69–75). The percentage of patients assessing their average pain in the last week as mild was 34% (95% CI: 30–36), as moderate was 56% (95% CI: 53–57) and as severe was 10% (95% CI: 8–12). Pain occurred in 39% (95% CI: 33–42) of patients in one place, 31% (95% CI: 28–36) in two places and 30% (95% CI: 29–31) in three or more places. Pain was located in the back (28%, 95% CI: 21–33), abdomen (16%, 95% CI: 14–19), arms (11%, 95% CI: 6–19), hips (19%, 95% CI: 16–22), knees (9%, 95% CI: 6–12), chest (18%, 95% CI: 16–19), feet (8%, 95% CI: 6–9), neck (7%, 95% CI: 6–9), and elbows or hands (9%, 95% CI: 6–10), and generalized pain occurred in 14% (95% CI: 8–16) of patients. According to the results of a simple analysis of the frequency and type of ailments, there was a statistically significant relation to the duration of illness. Patients suffering for more than 3 years more often experienced fatigue and pain, while patients with a shorter history of illness more often experienced constipation and lack of appetite. The differences were statistically significant (*p* = 0.03). As disease progressed, the number of symptoms reported by patients increased and their health assessment worsened. Using the Pearson test, the relations between the following variables were checked: disease duration and the number of reported symptoms (r = 0.08; M = 4.1), and disease duration and health self-assessment (r = 0.01). The obtained values did not confirm the relationship between the adopted variables.

#### 3.1.4. Performance Status

In the subjective assessment of fitness, after using the Karnofsky fitness index, it was shown that 28% (95% CI: 27–30) of patients declared the ability to perform normal physical activity, 45% (95% CI: 43–47) were able to take care of themselves, 49% (95% CI: 48–50) required periodic assistance, and 13% (95% CI: 8–16) required care and assistance (Table 3). Moreover, during the last month, 59% (95% CI: 58–60) of patients had to reduce their regular activity and 41% (95% CI: 38–45) reported that they were unable to work for medical reasons. Patients with a shorter history of illness more often reported incapacity to work (*p* = 0.01).

No connection between the type of cancer of the subjects and the occurrence of restrictions in everyday basic activities was confirmed. A strong correlation between the occurrence of restrictions in everyday basic activities and the number of different chemotherapy cycles was confirmed (Table 4).

#### 3.1.5. Impact of Duration of Disease

ESAS scores varied depending on the patient group in terms of activity, appetite and somnolence. A measurement effect was observed in all ESAS positions; somnolence worsened (*p* = 0.011) in patients with longer illness and remained stable in patients with shorter disease history. Appetite improved among patients who had been ill for a longer period compared to a shorter period (*p* = 0.025; *p* = 0.033). Activity assessed by ESAS improved among patients with longer disease history (*p* < 0.0001).

When asked about their self-esteem and quality of life, the vast majority of patients assessed them as average (67%, 95% CI: 64–69). In the assessment of the profile, quality of life and psychometric properties of EQ-5D-5L, it was shown that patients had the most severe problems in terms of self-care (81%, 95% CI: 76–89) and feeling anxious and depressed (63%, 95% CI: 60–68). A lower percentage of no problems in all 5 dimensions of EQ-5D-5L was found in patients with longer disease history. The mean of a single EQ-5D-5L index was 0.65 (95% CI: 0.63–0.67). Sociodemographic characteristics of the patients were negatively correlated with a single EQ-5D-5L index, while in patients with longer disease history, a single EQ-5D-5L index was stable and significantly increased (Table 5). In the course of statistical analyses, it was noticed that the duration of disease significantly influenced the assessment of the quality of life of the studied patients (*p* = 0.01), as opposed to their age (*p* = 0.53).

A strong correlation was found between quality of life and number of chemotherapy cycles. Nevertheless, a significant difference was found between the levels of quality of life in patients with ≤2 cycles and/or with 3–5 cycles (*p* < 0.001). It was also the case for the level of quality of life in patients with 6 cycles (*p* < 0.001) (Table 6).

## 4. Discussion

The World Health Organization defines quality of life as an individual’s perception of their position in life, in the context of their value systems and culture and concerning their goals, expectations and interests [14,15,16]. According to Siegrist and Jung, quality of life includes three closely related elements: physical indicators, mental determinants and social indicators. The quality of life of a specific person is always related to aspects that are significantly important to them [17,18]. Assessment of the quality of life of cancer patients takes up more and more space in the literature or discussions of specialists; moreover, it is becoming a standard. It is related to an individual, subjective approach to the patient and allows for the assessment of the impact of disease and treatment on functioning of the patient and their relatives in terms of physical, mental and social well-being. The purposefulness of measuring the quality of life in cancer patients was demonstrated by Montazieri, who stated that the global quality of life of patients before starting oncological treatment is an important predictor of survival [19,20,21]. Similarly, Li et al., based on a sample of over 400 cancer patients, presented results proving that health-related quality of life is a strong and independent predictor of overall survival [22]. As reported by Smyth EN and Jacob J., patients with advanced cancer often experience low quality of life caused by their illness and side effects of treatment. Health-related quality of life is a multifaceted well-being concept and is considered a priority area by oncologists [23,24].

As demonstrated by numerous studies, quality of life parameters deteriorate as a result of cancer diagnosis. Diagnosis of neoplastic disease usually causes severe anxiety, a sense of danger and insecurity, and often depression. These reactions stem from the social perception of cancer as a painful disease that is inevitably fatal [16,20]. The research carried out for this study shows that shock was a reaction to news about the disease in 38% of patients while 22% experienced a breakdown. The disease caused anger in 35% of patients, shame in 12% and depression in 30%. As many as 48% of respondents felt fear for their future. Similar results were obtained by Dehkordi et al., who showed that the most common problems among cancer patients treated with chemotherapy were fear of the future (29%), thinking about the disease and its consequences (26.5%), impatience (24%) and depression (17.5%) [25]. A study by Nayak MG et al., including a number of participants very similar to the number of subjects in our own research, showed that the mental well-being of the respondents influenced feelings of significant depression among 54.4% of participants and that the majority (98.3%) did not feel comfortable taking part in social life. Most of the patients feared disease recurrence (76.2%) and disability (62.1%) [26]. A comprehensive study of a US oncology centre involving nearly 4500 patients aged 19 years and older showed high rates of psychological symptoms that met the criteria for clinical diagnoses such as depression, adaptive disorders and anxiety, which was confirmed by subsequent studies by other scientists [27,28,29]. According to Charmaz and Stanton et al., chronic illness can cause guilt, loss of control, anger, sadness and confusion. Patients may also experience more general worries, such as fear of the future, inability to plan, and fear of changing sexual functions and changing role in the family [30,31]. In the studies by Mziray and Żuralska, all oncological patients experienced anxiety and the largest group of patients (45.8%) had moderate anxiety, all patients studied by the author and her team showed symptoms of depression and 45.8% had moderate depression [32]. The research of Kędra and Wiśniewski also showed that almost half (45.71%) of the respondents felt more anxious, sad and depressed [33]. Modlińska et al. showed that the anxiety response to cancer significantly affects the quality of life of terminally ill patients under 65 years of age, and it is known that, in older patients, problems related to everyday existence play the most important role [34]. Mood disorders are associated with a reduction in the quality of life of cancer patients, which has been demonstrated in numerous studies on these issues [35,36]. It is assumed that the prevalence of these abnormalities in the population of oncological patients is around 40% [20].

An extremely important aspect of the quality of life of cancer patients is the impact of the disease on their marital, family and social relations as well as on received support. The authors’ own research showed that, for the vast majority of respondents (37%), relations with their families and friends were not changed by the disease and remained satisfying while relations with their partners changed for the better in 28% of patients. During their illness, patients received support mostly from their spouses (62%). Gangane N. et al. obtained a very similar result using the same research tool and demonstrated that the lack of a partner was negatively related to quality of life, mental health and social relations [37]. Completely different results were obtained by Jacob J. et al.—in this research, unmarried patients reported higher social/family well-being compared to married patients and married women reported lower social/family well-being than unmarried women [24]. Unfortunately, as shown by the results of studies by Nayak MG et al., as many as 92.7% of oncological patients undergoing chemotherapy did not receive any support from friends and relatives [26].

Quality of life largely depends on the state of health, i.e., the impact of disease and treatment on the patient’s physical functioning. Cancer has a versatile impact on the lives of those affected, especially during chemotherapy treatment. It causes a decrease in the patient’s physical activity and influences a change in the appearance and a loss of the sense of attractiveness, which in turn reduces patient’s self-esteem [33]. Many authors emphasize the dependence of the quality of life of cancer patients on the applied anticancer therapy. In the study by Słowik-Gabryelska [38], in more than half of patients with primary lung cancer, it was observed that the side effects of chemotherapy decreased overall performance and had a negative impact on quality of life. A significant relation between quality of life and line of treatment is also reported. Along with the next line of chemotherapy, the quality of life of the studied patients deteriorated significantly [39]. The conclusions of Zielińska-Więczkowska B. from the study conducted using the EORTC QLQ-C30 questionnaire are very interesting, showing that treatment with cytostatics reduces the quality of life of patients but to an extent that is not statistically significant [40]. In the author’s own research, the vast majority of patients assessed their self-esteem and quality of life as average. The assessment of the profile, quality of life and psychometric properties of EQ-5D-5L showed that the patients had the most severe problems in terms of self-care and feeling anxious and depressed. The result is comparable to the study conducted by Adamowicz K. and Waliszewska Z., and the similarity may result from the study design and the tool used. After 6 months of therapy, the majority of respondents (66%) stated that they did not feel tired and 23% of respondents felt a clear lack of energy. Patients most often complained of weakness (80%). Many patients reported nausea and vomiting (60%). Among the limitations in everyday life resulting from treatment, the majority of patients (58%) indicated inability to continue working. The vast majority of respondents (88%) described their health condition as bad, and the disease limited the social activity of 70% of patients. In the study population, the mean general quality of life before treatment was 60.92 and, after treatment, was 58.20 [41]. The results of this study are supported by Gandhi et al. [42], who conducted a cohort study of 100 patients suffering from multiple symptoms such as pain, insomnia, loss of appetite and fatigue. These symptoms adversely affected the normal functioning of the patients. Emotional functioning deteriorated in 50% of patients, and physical functioning deteriorated in nearly 23% of the remaining 50% of the population. A study by Kannan et al. also showed that the overall mean Quality of Life (QoL) result of the study population was 122.38 ± 13.86, and the mean of approximately 80% of the population was below the mean QoL [37]. Similar results were observed in the study by Nayak MG et al. [26]. The results of other scientific studies also show a significant reduction in QoL due to typical symptoms resulting from cancer [43,44,45]. Many authors have reported that treatment side effects affect a patient’s QoL depending on individual circumstances, type of cancer and its treatment [26].

Out of the somatic ailments that significantly modify the quality of life of oncological patients, fatigue is the most emphasized in literature. Many researchers analysing the quality of life of cancer patients undergoing anticancer therapy emphasize the prevalence of this symptom in the studied patients. Currently, it is believed that fatigue is the most frequently reported symptom of cancer and the therapy used; it affects approximately 80% of cancer patients and 70% of those treated. It appears often before diagnosis, very often as an initial symptom of the disease, and its severity does not decrease even after rest. Chronic cancer fatigue may persist for months or even years after the end of cancer treatment [16]. In the studies by Kieszkowska-Grudny A. et al., symptoms of chronic fatigue were reported by the vast majority of respondents (72%) [46], according to Kapela et al., by 51% of patients [16]. Similarly, studies by Smets EMA et al. showed that, among half of the patients treated, oncological fatigue lasted up to 3 months after the end of therapy and, in 20%, it was of high intensity [47]. The results of this study also indicate that fatigue is the most troublesome symptom of cancer, which is declared by more than half of the surveyed patients (76%). The other most burdensome symptoms mentioned by the patients were lack of appetite (71%), difficulty sleeping (38%) and constipation (65%). This is confirmed by the studies by Kapela et al., which showed that other ailments reported by the respondents were nausea (20.7%), pain (20.7%) and lack of appetite (16.3%) [16]. Sleep disorders are confirmed by the studies by Kaczmarek-Borowska B. [48], Zielińska-Więczkowska B. [40] and Nowicki A. [49]. A significant modifier of quality of life is pain experienced by patients, which is confirmed by our own research and by Thielking PD [50] and Wool MS, Mor V. [51]. According to the research of Kroenke K, et al., of 405 participants, 24% experienced only pain and 44% experienced both depression and pain [52]. Research by Nayak MG et al., conducted with the use of similar tools and a similar sample, showed that the low physical well-being of the respondents was influenced by pain (72.9%), sleep problems (71.7%) and fatigue (91.8%) [26]. Data from the National Health Interview Survey (NHIS) show that cancer at least doubles the probability of ill health and disability [1]. Physical impairments and disabilities as well as the fatigue and pain experienced by cancer patients often lead to an inability to perform routine activities of everyday life. According to the research of Yabroff et al. in the United States, adults with an early diagnosis of cancer report the need for help with daily activities more often than people of similar age, sex and level of education without such diagnosis [53]. The data of the National Health Interview Survey show that people who survived cancer without any other chronic disease reported twice as often a reduced ability to perform daily activities than people without a history of cancer or another chronic disease [54]. In the author’s research, the subjective assessment of fitness showed that 32% requires periodic help and 10% requires care and assistance. Moreover, during the last month, 59% of patients had to reduce their regular activity and 41% reported that they were unable to work for medical reasons. As shown by numerous studies, in addition to deterioration in the quality of life and functional status, physical discomfort, especially pain, is the most common cause of decreased work performance and causes a significant degree of total disability [55]. In the studies by Kroenke K. et al., patients reported more than 60% of days in bed or a significant reduction in activity and a health-related unemployment rate of 43% [52]. Also, the meta-analysis by de Boer A. et al. showed that the unemployment rate in cancer patients is more than twice as high as in the control group (34% vs. 15%) [56]. Very interesting results were obtained by Jacob J. et al., who used very similar research tools and showed that patients with higher results of financial difficulties reported worse functional well-being, worse emotional well-being, a lower subscale of spiritual well-being and higher symptoms of anxiety and depression [24]. A review of the study results clearly shows that, with time, the quality of life of patients gradually deteriorates [57,58]. A regression in occupational functioning was observed, and it persisted even after the end of treatment. Restriction in taking up professional activity in cancer patients lowered their quality of life [59].

In summary, the quality of life of patients treated with chemotherapy deteriorates, especially due to suffering, sadness, anxiety, lack of vital energy, fear of treatment and accompanying physical ailments. Patients struggle with emotional, social and personal problems and very often have to accept changes in their appearance. The results of this study showed that there was no correlation between the quality of life and age, gender, social status, marriage and work. Similar results have been reported by Vedat et al. and Rustøen. Moreover, a correlation was found between the degree of disease and quality of life. Also, Rustøen and Holzner in two separate studies found that quality of life was related to time, noting a decrease in the quality of life of cancer patients with an increase in the extent of the disease [16,60].

Our qualitative study included patients with different types of cancer. These points make our results transferable to other cancer contexts. However, there are some limitations to consider. Firstly, the study was conducted in Poland and our results need to be cautiously transferred elsewhere, as oncological care in Poland is highly dependent on the organization of the medical system and the economy of the country. Secondly, we recruited patients diagnosed during the past 5 years. We took sufficient time to assist the patients and patient advocates in recalling their long-term memory and to minimize the effect of recall bias. The sample was too small for subgroup analysis. Our study should be interpreted as exploratory because there are no reliable data available.

## 5. Conclusions

Cancer undoubtedly has a negative impact on the quality of life of patients, which is related to the disease process itself, the treatment used and the duration of the disease. The necessity of frequent hospitalizations, negative emotions and numerous somatic ailments that change over time significantly reduce the quality of life of cancer patients.Somatic symptoms accompany patients at every stage of the disease and are associated with increased disability and reduced quality of life. The factors that significantly influence the occurrence of symptoms depend on the phase of the disease, the cycles of chemotherapy and the duration of the disease.To achieve the best possible quality of life despite disease, it is important to regularly assess the quality of life of patients to quickly assess the problems of each sphere of life, which will enable the identification of high-risk patients and allow for early intervention depending on the identified needs or deficits. Undetected and untreated disorders threaten the results of cancer therapies, reduce the quality of life of patients and increase healthcare costs.

## Figures and Tables

**Table 1 ijerph-17-06938-t001:** Descriptive statistics of the examined group of patients.

Demographic Data	Total
	*N* = 800
**Sex**	
women (*N*/%)	480/60%
men (*N*/%)	320/40%
**The Age of the Study Group**	
SD	54.53 (8.86)
95% CI	<26; 75>
**Place of Residence**	
city (*N*/%)	336/42%
village (*N*/%)	464/58%
**Financial Situation**	
very good (*N*/%)	80/10%
good (*N*/%)	344/43%
average (*N*/%)	280/35%
bed (*N*/%)	96/12%
**Age Groups**	
39–41 (*N*/%)	120/15%
42–52 (*N*/%)	200/25%
53–63 (*N*/%)	296/37%
64–75 (*N*/%)	184/23%
**Education of the Study Group**	
higher education (*N*/%)	64/8%
secondary education (*N*/%)	264/33%
vocational education (*N*/%)	360/45%
primary education (*N*/%)	112/14%
**Marital Status**	
married (*N*/%)	512/64%
widowed (*N*/%)	184/23%
unmarried (*N*/%)	104/13%
**Source of Income**	
professionally active (*N*/%)	584/73%
annuity (*N*/%)	160/20%
retirement (*N*/%)	56/7%
**Times of Illness**	
3–12 m. (*N*/%)	184/23%
1–2 y. (*N*/%)	136/17%
3–5 y. (*N*/%)	480/60%
**Type of Cancer**	
breast (*N*/%)	240/30%
uterine (*N*/%)	192/24%
lung (*N*/%)	160/20%
colorectal (*N*/%)	160/20%
bone(*N*/%)	48/6%
**Number of Chemotherapy Cycles**	
2≤ (*N*/%)	224/28%
3–5 (*N*/%)	344/43%
≤6 (*N*/%)	232/29%

**Table 2 ijerph-17-06938-t002:** Symptoms checklist.

Symptoms	Times of Illness			*p*	Sex		*p*
	3–12 Months	1–2 Years	3–5 Years		Women	Men	
Characteristics N/%							
Fatigue	33/18%	41/30%	365/76%	0.03	107/22%	144/45%	0.02
Pain	61/33%	23/17%	288/60%	0.03	389/81%	234/73%	0.41
Change in skin condition	83/45%	54/40%	197/41%	0.88	168/35%	211/66%	0.01
Weight loss	64/35%	37/27%	125/26%	0.54	187/39%	122/38%	0.91
Loss of appetite	131/71%	34/25%	139/29%	0.03	346/72%	250/78%	0.88
Nausea	86/47%	35/26%	86/18%	0.35	216/45%	154/48%	0.54
Vomiting	86/47%	30/22%	82/17%	0.88	235/49%	186/58%	0.91
Constipation	120/65%	20/15%	120/25%	0.03	317/66%	195/61%	0.44
Diarrhea	46/25%	16/12%	96/20%	0.54	91/19%	58/18%	0.91
Abdominal pains	101/55%	29/21%	91/19%	0.55	235/49%	186/58%	0.55
Headaches	64/35%	20/15%	48/10%	0.88	91/19%	58/18%	0.91
Dizziness	57/31%	39/29%	86/18%	0.88	130/27%	106/33%	0.55
Loss of hair	156/85%	88/65%	216/45%	0.41	427/89%	282/88%	0.44
Depression	101/55%	44/32%	322/67%	0.54	139/29%	186/58%	0.91
Difficulty sleeping	107/58%	23/17%	106/22%	0.54	91/19%	58/18%	0.71
Anxiety	105/57%	48/35%	331/69%	0.44	149/31%	154/48%	0.91
Difficulty concentrating	15/8%	10/7%	106/22%	0.09	101/21%	93/29%	0.74
Fear for the future	53/29%	50/37%	379/79%	0.01	168/35%	243/76%	0.01

**Table 3 ijerph-17-06938-t003:** The Karnofsky performance status.

Degree of Efficiency	Description	Duration of Illness				*p*
		3–12 Months	1–2 Years	3–5 Years	Together	
		Characteristics *N*/%				
100	Normal condition, no complaints or symptoms	10/5%	14/10%	24/5%	48/6%	0.41
90	State of normal activity, slight complaints and symptoms of the disease	36/20%	34/25%	154/32%	224/28%	0.01
80	Almost active state (requires some effort); slight complaints and symptoms of the disease	2/1%	14/10%	48/10%	64/8%	0.41
70	State of inability to perform work or proper activity, with the ability to self-service	40/22%	42/31%	278/58%	360/45%	0.01
60	Condition requiring periodic care, while preserved the ability to independently fulfill most of your daily needs	46/25%	49/31%	297/62%	392/49%	0.01
50	A condition that requires frequent care and frequent medical interventions	10/5%	14/10%	48/10%	72/9%	0.41
40	State of failure and need for special care	2/1%	2/1%	100/21%	104/13%	0.41
30	State of severe insufficiency, indications for hospitalization	0/0%	0/0%	24/5%	24/3%	0.41
20	Serious illness, absolute necessity of hospitalization and providing supportive care	0/0%	0/0%	24/5%	24/3%	0.41
10	The state of sudden increase in the threat to life	0/0%	0/0%	24/5%	24/3%	0.41
0	Death	0/0%	0/0%	0/0%	0/0%	0

**Table 4 ijerph-17-06938-t004:** Impact of physical ailments related to the disease on limitations in everyday basic activities.

Symptoms		Number of Chemotherapy Cycles			*p*	Type of Cancer					*p*
		2≤	3–5	≤6		Breast	Uterine	Lung	Colorectal	Bone	
		Characteristics *N*/%									
Impact of illness symptoms on restrictions in daily activities	no limit	29/ 13%	45/ 13%	35/ 15%	0.001	161/67%	165/86%	113/71%	122/76%	33/69%	0.994
	partly	107/ 48%	265/ 77%	172/ 74%		58/24%	10/5%	29/18%	32/20%	11/23%	
	completely	88/ 39%	34/ 10%	25/ 11%		21/9%	17/9%	18/11%	6/4%	4/8%	

**Table 5 ijerph-17-06938-t005:** EQ-5D-5L profile of patients by different disease stages.

Duration of Illness	3–12 Months	1–2 Years	3–5 Years
Questionnaire EQ-5D-5L	Characteristics *N*/%		
**Mobility**			
No problems	107/58%	73/54%	250/52%
Slight problems	37/20%	39/29%	129/27%
Moderate problems	27/15%	14/10%	43/9%
Severe problems	10/5%	7/5%	29/6%
Unable to walk about	3/2%	3/2%	29/6%
**Self-Care**			
No problems	164/89%	109/80%	360/75%
Slight problems	13/7%	16/12%	58/12%
Moderate problems	2/1%	6/4%	24/5%
Severe problems	3/2%	4/3%	14/3%
Unable to wash or dress myself	2/1%	1/1%	24/5%
**Usual Activities**			
No problems	138/75%	87/64%	293/61%
Slight problems	30/16%	31/23%	100/21%
Moderate problems	8/4%	10/7%	29/6%
Severe problems	5/3%	4/3%	24/5%
Unable to do	3/2%	4/3%	34/7%
**Pain/Discomfort**			
No problems	103/56%	50/37%	202/42%
Slight problems	60/33%	63/47%	182/38%
Moderate problems	8/4%	14/10%	48/10%
Severe problems	10/5%	6/4%	24/5%
Extreme pain	3/2%	3/2%	24/5%
**Anxiety/Depression47**			
Not anxious or depressed	70/38%	31/23%	139/29%
Slightly	71/39%	60/44%	197/41%
Moderately	25/14%	25/19%	72/15%
Slightly	10/5%	14/10%	38/8%
Extremely	8/4%	6/4%	34/7%

**Table 6 ijerph-17-06938-t006:** Impact of the number of chemotherapy cycles on the quality of life.

Number of Chemotherapy Cycles	Quality of Life			*p*
	Non-Favourable	Fairly Favourable	Favourable	
	Characteristics *N*/%			
2≤	36/16%	150/67%	38/17%	0.001
3–5	34/10%	265/77%	45/13%	0.001
≤6	19/8%	116/50%	97/42%	0.001

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
