# Peer review of "Quality of Life of Cancer Patients Treated with Chemotherapy"

_ijerph, 2020, doi:10.3390/ijerph17196938_

Round 1

Reviewer 1 Report

Although the manuscript focuses on a very topical subject, the quality of life during chemotherapy treatments, the literature reported is not recent while a recent exhaustive literature exists. In addition, the purpose of the study focuses on the trajectory of treatment care when more than 60% of participants are in the period of survivorship care (3-5 years).

Methodology

Lack of clarity on the criteria for inclusion and exclusion.

References are missing about the measurement tools used.

Although the authors argue that the questionnaire for the qualitative interview has acceptable levels of internal consistency and test-retest reliability and construct validity, this data is not presented to the reader.

Results and discussion

As the purpose of the study, which looks at the treatment period and that 60% of the participants are no longer in this phase of the care trajectory, the results should be interpreted with caution.

Author Response

Dear Reviewer,

We would like to thank the reviewers for their comments. After analysing all the comments, we made the following changes:

Although the manuscript focuses on a very topical subject, the quality of life during chemotherapy treatments, the literature reported is not recent while a recent exhaustive literature exists. In addition, the purpose of the study focuses on the trajectory of treatment care when more than 60% of participants are in the period of survivorship care (3-5 years).

  1. Latest literature added as recommended (line 496-610).
  2. The study included patients undergoing chemotherapy, but with different chemotherapy cycles; we added the relevant results. In the next study, as suggested, we will separate subjects during maintenance treatment and during lifetime treatment (line 160-161; 270-275; 293-298).

Lack of clarity on the criteria for inclusion and exclusion.

  1. In the Methods section, criteria for the recruitment, inclusion and exclusion of participants have been added (line 67-98).

References are missing about the measurement tools used.

  1. In the Discussion part, the results were compared with other similar studies (line 299-447).

Although the authors argue that the questionnaire for the qualitative interview has acceptable levels of internal consistency and test-retest reliability and construct validity, this data is not presented to the reader.

  1. We apologize for the editorial error. The method used in the study was a clinical, direct, individual, structured interview, which was in-depth and focused. The qualitative interview questionnaire was a standardized measuring instrument and was verified by testing a group of 30 patients during the month (line 106-107).

As the purpose of the study, which looks at the treatment period and that 60% of the participants are no longer in this phase of the care trajectory, the results should be interpreted with caution.

  1. In the Results section, a table with the characteristics of the research group and the impact of the type of cancer and chemotherapy cycles on the quality of life has been added (line 160-191; 270-275; 293-298).
  2. In the Discussion section, information about the limitations of the study has been added (line 448-455).

I hope that the changes made are satisfactory and this will allow publication. I am asking you to take into account the positive comments of the reviewers that this is an interesting study and a good study.

Sincerely

Reviewer 2 Report

The article is overall well written and reads well. As a physician managing patients with cancer, we see patients undergoing extreme emotional stress that does change them in enumerable ways. The work of authors attempts to assess the impact of cancer diagnosis on the various aspects of quality of life. This is a difficult task, as many of the factors are indeed subjective. The authors did well to incorporate various questionnaires to best capture these factors. The authors performed a population-based cross-sectional study which is appropriate. The inclusion and exclusion criteria are appropriate. The authors found a significant impact of cancer on quality of life as expected and interesting the impact was greater in patients who had a diagnosis for 3 or more years. Would recommend including some limitations of the study which may include but are not limited to the cross-section nature of the study. Quality would also depend on the type of cancer as well as the stage of the disease as a diagnosis. Overall, the study does capture the tremendous psychological and physical impact of a cancer diagnosis on the life of patients and remains of interest to the readers. Thank you for the opportunity to review your work.

Strength:
The topic is of interest.
Study design is appropriate.
Inclusion and exclusion criteria are acceptable.

Weakness/ suggestion:
The results could be presented in more organized manner to make it more easily readable. Suggestion for topics would include
• Baseline demographics
• Psychological Impact – questionnaire and wellbeing
• Impact on social interactions.
• Symptoms
• Performance status
• Impact of duration of disease
For discussion should include limitation of the studies
Cross sectional nature means it captures in a given moment and does not correlate with how the quality of life changes over a period of time. This partly addressed in the study with attempting to assess impact of duration of diagnosis.
Moreover, quality of life can vary greatly between different cancers and types of chemotherapy. This is not within the scope of the study but should be acknowledged.

Author Response

Dear Reviewer,

We would like to thank the reviewers for their comments. After analysing all the comments, we made the following changes:

The results could be presented in more organized manner to make it more easily readable. Suggestion for topics would include: Baseline demographics, Psychological Impact – questionnaire and wellbeing, Impact on social interactions, Symptoms, Performance status, Impact of duration of disease

  1. The results are presented in a more structured way, divided into subsections: Baseline demographics, Psychological Impact – questionnaire and wellbeing, Impact on social interactions, Symptoms, Performance status, Impact of duration of disease (line line 156, 179, 210, 227, 261, 275).

For discussion should include limitation of the studies

  1. In the Discussion section, information about the limitations of the study has been added (line 448-455).

Cross sectional nature means it captures in a given moment and does not correlate with how the quality of life changes over a period of time. This partly addressed in the study with attempting to assess impact of duration of diagnosis.

  1. Latest literature added as recommended (line 496-610).
  2. The study included patients undergoing chemotherapy, but with different chemotherapy cycles; we added the relevant results. In the next study, as suggested, we will separate subjects during maintenance treatment and during lifetime treatment (line 160-191; 270-275; 293-298).

Moreover, quality of life can vary greatly between different cancers and types of chemotherapy. This is not within the scope of the study but should be acknowledged.

  1. In the Results section, a table with the characteristics of the research group and the impact of the type of cancer and chemotherapy cycles on the quality of life has been added (line 160-161; 270-275; 293-298).

I hope that the changes made are satisfactory and this will allow publication. I am asking you to take into account the positive comments of the reviewers that this is an interesting study and a good study.

Sincerely

Anna Lewandowska

Round 2

Reviewer 1 Report

The authors have enhanced the manuscript by adding recent literature and the results are better targeted.